# Comparative Analysis of *In Vivo* and *In Vitro* Virulence Among Foodborne and Clinical *Listeria monocytogenes* Strains

**DOI:** 10.3390/microorganisms13010191

**Published:** 2025-01-17

**Authors:** Hui Yan, Biyao Xu, Binru Gao, Yunyan Xu, Xuejuan Xia, Yue Ma, Xiaojie Qin, Qingli Dong, Takashi Hirata, Zhuosi Li

**Affiliations:** 1School of Health Science and Engineering, University of Shanghai for Science and Technology, Shanghai 200093, China; yanhui015527@163.com (H.Y.); gao_binru@163.com (B.G.); 2135070609@st.usst.edu.cn (Y.X.); xiaxj@usst.edu.cn (X.X.); yuema@usst.edu.cn (Y.M.); qxj@usst.edu.cn (X.Q.); dongqingli@126.com (Q.D.); 2Shanghai Municipal Center for Disease Control and Prevention, Shanghai 200051, China; xubiyao@scdc.sh.cn; 3Graduate School of Agriculture, Kyoto University, Kyoto 606-8501, Japan; hirata.takashi.a03@kyoto-u.jp; 4Faculty of Rehabilitation, Shijonawate Gakuen University, Osaka 574-0011, Japan

**Keywords:** *Listeria monocytogenes*, virulence, hemolysis activity, RT-qPCR, *Galleria mellonella*, JEG-3 cells

## Abstract

*Listeria monocytogenes* is one of the most important foodborne pathogens that can cause invasive listeriosis. In this study, the virulence levels of 26 strains of *L. monocytogenes* isolated from food and clinical samples in Shanghai, China, between 2020 and 2022 were analyzed. There were significant differences among isolates in terms of their mortality rate in *Galleria mellonella*, cytotoxicity to JEG-3 cells, hemolytic activity, and expression of important virulence genes. Compared with other STs, both the ST121 (food source) and ST1930 (clinic source) strains exhibited higher *G. mellonella* mortality. The 48 h mortality in *G. mellonella* of lineage II strains was significantly higher than that in lineage I. Compared with other STs, ST1930, ST3, ST5, and ST1032 exhibited higher cytotoxicity to JEG-3 cells. Based on the classification of sources (food and clinical strains) and serogroups (II a, II b, and II c), there were no significant differences observed in terms of *G. mellonella* mortality, cytotoxicity, and hemolytic activity. In addition, ST121 exhibited significantly higher *hly*, *inlA*, *inlB*, *prfA*, *plcA*, and *plcB* gene expression compared with other STs. A gray relation analysis showed a high correlation between the toxicity of *G. mellonella* and the expression of the *hly* and *inlB* genes; in addition, *L. monocytogenes* may have a consistent virulence mechanism involving hemolysis activity and cytotoxicity. Through the integration of *in vivo* and *in vitro* infection models with information on the expression of virulence factor genes, the differences in virulence between strains or subtypes can be better understood.

## 1. Introduction

*Listeria monocytogenes*, a Gram-positive facultative anaerobic bacterium, can cause listeriosis in humans [1]. It primarily affects vulnerable populations such as newborns, elderly individuals, and those with compromised immune systems [2]. The main manifestations after infection include gastroenteritis, septicemia, meningitis, and mononucleosis [1]. Outbreaks of human listeriosis are closely associated with contamination with *L. monocytogenes* in various food categories, including ready-to-eat foods, dairy products, aquatic products, and meat products [3,4]. Although the contamination rate of this bacterium is not high compared with that of other foodborne pathogens, the mortality rate among susceptible individuals after infection is higher (20–30%) [5].

*L. monocytogenes* isolated from different environments (clinical, environmental, and food) may exhibit variability in growth, resistance, and virulence. This phenomenon may be attributed to the ability of *L. monocytogenes* to dynamically regulate gene expression in response to diverse environmental conditions [6]. *L. monocytogenes* experiences a complex *in vivo* environment upon ingestion and has the ability to survive even after passing through the gastrointestinal tract [7]. Numerous studies have demonstrated that when *L. monocytogenes* colonizes the intestinal environment, the expression levels of its related virulence genes change, and it can even reshape its entire transcriptome, thereby altering its own pathogenicity [8,9,10]. Møretrø et al. [11] demonstrated that *L. monocytogenes* strains isolated from human clinical samples were more virulent than those isolated from food; however, there may also be national or regional differences in this result.

*L. monocytogenes* could be classified into multiple types based on different classification methods. The characterization and differentiation of *L. monocytogenes* at the strain level are accomplished through molecular methods such as multi-locus sequence typing (ST) [12]. The 13 serotypes of *L. monocytogenes* are divided into four major genetic diversity lineages: lineages I, II, III, and IV [1]. Lineages I and II contain clinically relevant strains, as noted in major reports on listeriosis epidemics, while lineage II isolates or serotypes are more commonly found in food [1]. Additionally, the serotypes are also classified into molecular PCR-based serotypes: II a (including serotypes 1/2a and 3a), IIb (including 1/2b and 3b), II c (including 1/2c and 3c), and IV b (including 4b, 4d, and 4e) [13]. Among these, three serotypes (1/2a, 1/2b, and 4b) account for over 95% of invasive listeriosis cases [12,14]. Although all strains of *L. monocytogenes* have been clinically reported to be virulent, the level of virulence and the presence of pathogenic genes in this pathogen exhibit high genetic heterogeneity [15]. The potential determinants of *L. monocytogenes* virulence depend on the differences in the strain isolation source, serotype, lineage, and ST [1]. Myintzaw et al. confirmed that the virulence variations among different strains of *L. monocytogenes* were associated with the strain’s ST [16]. Furthermore, some studies have also indicated that serotype and lineage could serve as key factors in distinguishing the virulence levels of *L. monocytogenes* strains [17,18].

The prevalence of different subtypes of *L. monocytogenes* varies significantly across countries and regions. For instance, serotype 1/2a (II a), 1/2b (IIb), and 4b (IV b) strains have been associated with outbreaks of listeriosis in the United States [19]; however, the most prevalent serotypes in Italy and China are II a, II b, and II c [20,21]. In addition, in Japan, the most common strains of *L. monocytogenes* belong to ST6, followed by ST1 and ST2, while strains of ST1, ST3, ST4, and ST6 are strongly associated with listeriosis cases in European countries [22,23]. In food products illegally imported into the EU, *L. monocytogenes* strains belonging to ST9 and ST121 have been reported to exhibit reduced virulence potential compared to strains of the same STs in the source region, indicating that strain variability across different regions can lead to changes in virulence levels [24]. Currently, most of the risk assessment work and studies on the pathogenic mechanisms of *L. monocytogenes* rely on standard strains, which can result in an overestimation or underestimation of the risks posed by the pathogenic bacteria. Therefore, it is crucial to conduct a differential analysis of the characteristics of different subtypes of isolates in specific regions. In our previous work, we have conducted preliminary explorations of the differences in growth, biofilm formation, and environmental resistance among strains of different subtypes [2,25]. In this study, we focus on the heterogeneity of strains of different subtypes in terms of virulence.

In this study, 26 strains of *L. monocytogenes* from different sources (food and clinical) were studied in Shanghai, China, from 2020 to 2022. Among these, 12 clinical strains were isolated from listeriosis cases in six sentinel hospitals (including two large obstetrics and gynecology specialist hospitals, two pediatric specialist hospitals, and one general hospital) in collaboration with the Shanghai Center for Disease Control (SCDC) during the period of 2020 to 2022. An additional 14 strains of *L. monocytogenes* were isolated from high-risk foods such as raw meat, cooked meat products, and processed meat products between the years of 2020 and 2022. This study focuses on *L. monocytogenes* strains and investigates their potential pathogenicity as well as the expression levels of virulence factors. The aim is to assess the correlation between strain virulence levels and subtypes, providing a scientific basis for an understanding of the epidemiological characteristics of *L. monocytogenes* in food and clinical settings, and facilitating the development of targeted prevention and control measures.

## 2. Materials and Methods

### 2.1. Bacterial Strains and Preparation of Bacterial Suspension

From 2020 to 2022, a total of 26 *L. monocytogenes* strains were isolated in Shanghai, China. Among them, 12 were from clinical sources (112–123), and the remaining 14 were obtained from food samples (124–137). According to SNP evolutionary analysis, the SNPs of strains 117 and 118 are completely identical, whereas there are substantial SNP differences among the other strains [25]. There is no traceability relationship between clinical and food source strains [25]. The reference strain of *L. monocytogenes* ATCC 19112 (ST122) was obtained from the China Industrial Culture Preservation Center (CICC, Beijing, China, http://www.china-cicc.org/ (accessed on 23 October 2024)). All frozen strains of *L. monocytogenes* were preserved in 50% glycerol and trypticase soy broth with 0.6% yeast extract (TSB-TE; Beijing Road and Bridge Technology Co., Ltd., Beijing, China) at a temperature of −80 °C. *Listeria monocytogenes* was inoculated on trypticase soy agar with 0.6% yeast extract (TSA-YE; Beijing Road and Bridge Technology Co., Ltd., Beijing, China) and stored at 4 °C until utilization (within a maximum period of one month). To activate the strains prior to the start of the experiment, a single colony was inoculated into 10.0 mL TSB-YE and incubated at 37 °C for 18–24 h.

### 2.2. Assessing the Pathogenicity of L. monocytogenes Isolates Against Galleria mellonella

The pathogenicity of *L. monocytogenes* was assessed using the *G. mellonella* model as described by Martinez et al. [26]. Briefly, *G. mellonella* larvae (6 weeks old, with a milky white color, and measuring 2–3 cm in length) were selected. The activated culture of bacteria was diluted and resuspended with phosphate-buffered saline (PBS) to prepare a bacterial suspension with a concentration of 1 × 10^6^ CFU/mL. The 10 μL prepared bacterial suspension (10^4^ CFU per larva) was then injected into the larvae of *G. mellonella* (*n* = 30 per treatment group) through the blood sacs using a syringe with a pinhole diameter of 0.26 mm. An equal amount of PBS was injected into members of a negative control group, while the suspension of strain ATCC 19112 was injected into members of a positive control group. The larvae were incubated at a constant temperature of 37 °C with controlled humidity in a dark environment. *G. mellonella* survival was observed at 6, 12, 24, 48, 72, 96, and 120 h after infection without requiring feeding.

### 2.3. Cell Culture

Human placental choriocarcinoma (JEG-3) cells (CM-H072, generation 10–20) were obtained from Gaining Biology (Shanghai, China). They were cultured in a Minimum Essential Medium (MEM; Gibco, Invitrogen, Carlsbad, CA, USA) containing 10% fetal bovine serum (FBS, HyClone Laboratories, Logan, UT, USA) and 1% penicillin–streptomycin solution (Hyclone) [27]. Briefly, the cells were inoculated at a density of 4–5 × 10^4^ cells/cm^2^ and cultured for 1–2 d with daily medium changes. When the cells reached 80% fusion, the cells were detached from the culture dish using trypsin (gibco, 0.25%) for 3 min at 37 °C. The cells were resuspended in a fresh medium and cultured in a humidified incubator at 37 °C with 5% CO_2_.

### 2.4. Cytotoxicity Assays

The survival rate of JEG-3 cells was determined as a cytotoxicity index using the cell counting Kit-8 (cck-8; Langeke Technology Co., Ltd., Beijing, China) described by Wang et al. [28], with some modifications. Briefly, approximately 4000 cells were seeded into each well of a 96-well plate and incubated at 37 °C for 48 h. Then, the strains were incubated for an additional 24 h at 37 °C. The activated *L. monocytogenes* culture was centrifuged at 8000× *g* for 1 min, resuspended in an MEM, and then diluted to a final bacterial suspension with a concentration of 1 × 10^6^ CFU/mL. A total of 20 μL of diluted bacterial solution was added to each well and co-cultured with the cells for 24 h. Also, 10 μL of cck-8 was added to each well and incubated at 37 °C for 2 h before the absorbance value was measured at 450 nm (A450 nm) using a microplate reader (Read-Max1900, Sampo Biotech Ltd., Shanghai, China). The PBS was used as a blank control, and a *Listeria innocua* bacterial solution was used as a negative control. Each set of experiments was repeated 5 times. The percentage of cytotoxicity is calculated as follows:


$$Cytotoxicity\%=\left[1-\frac{A450\;tested\;sample-A450\;blank}{A450\;control\;sample-A450\;blank}\right]\times 100\%$$


### 2.5. Hemolysis Assay

Hemolytic activity was assessed using sheep erythrocytes (4%, R21900-100, Shanghai Yuanye Bio-technology Co., Ltd., Shanghai, China), as described by Li et al. [29]. The activated *L. monocytogenes* cultures (10^8^–10^9^ CFU/mL) were centrifuged at 5500× *g* for 10 min at 4 °C. Then, 250 μL of the supernatant was added to a mixture containing 500 μL of hemolysin buffer (S25861, Shanghai Yuanye Bio-technology Co., Ltd., Shanghai, China) and 250 μL of 4% sheep erythrocyte solution. Then, 250 μL of PBS as a negative control (0% hemolysis) or 250 μL of 1% (*v*/*v*) Triton X-100 as a positive control (100% hemolysis) was added to the hemolysin buffer and sheep erythrocyte solution. The samples were incubated at 37 °C for 30 min and then centrifuged at 5500× *g* for 10 min at 4 °C. The absorbance values of the supernatants were determined at 450 nm using a microplate reader (Read-Max1900, Sampo Biotech Ltd., Shanghai, China). The percent hemolysis was determined by comparing the sample with a positive control.

### 2.6. RT-qPCR Analysis

A 20 mL bacterial suspension (10^9^ CFU/mL) was centrifuged at 8000× *g* for 2 min to obtain the cell precipitation. The total RNA of each sample was extracted using the Bacterial Total RNA Extraction Kit (Genstone Biotech Co., Ltd., Beijing, China). cDNA was synthesized from RNA using the Hiscript^®^ II Reverse Transcriptase Kit (Vazyme Biotech Co., Ltd., Nanjing, China). The cDNA samples were stored at −20 °C until use. Each 20 μL RT-qPCR reaction system contained 10 μL of the 2xTaq Pro Universal SYBR qPCR Master Mix (Nanjing vazyme Biotech Co., Ltd., Nanjing, China), 0.4 μL of upstream primers, 0.4 μL of downstream primers (Table 1), 1.2 μL of template cDNA, and 8.0 μL of RNase-Free ddH_2_O. The reaction was performed using an RT-qPCR system (F00-96A, BIOER, Hangzhou, China), and 16S rRNA was used as an internal reference gene. The amplification program was as follows: 1 cycle at 95 °C for 30 s, 40 cycles at 95 °C for 5 s, and 40 cycles at 60 °C for 30 s; then, a melt curve program was carried out as follows: 95 °C for 15 s, 60 °C for 60 s, and 95 °C for 15 s. The relative expression level of genes was analyzed using the comparative threshold cycling (2^−ΔΔCT^) method [30].

### 2.7. Statistical Analysis

All experiments were performed in triplicate, and the results are presented as mean ± standard deviation (SD) (*n* = 3). Statistical analyses were performed using the statistical package SPSS 21.0 software (SPSS Inc., IBM Corporation, Armonk, NY, USA). GraphPad Prism 9.3 (GraphPad Software, San Diego, CA, USA) was used for plotting graphs. Data were compared for statistical significance using analysis of variance (ANOVA) techniques and Duncan’s multiple range test. The level of statistical significance was set at *p* < 0.05. 

## 3. Results

### 3.1. The Pathogenicity of Different L. monocytogenes Strains in the G. mellonella Model

The survival rate of *G. mellonella* larvae was investigated after 26 strains of *L. monocytogenes* were injected. The median lethal time (LT_50_) of different strains is listed in Table 2. The results showed that the LT_50_ of strains 112 (ST87), 114 (ST1930), 127 (ST121), and 128 (ST121) was less than 12 h; in addition, the LT_50_ of strains 115, 125, and 136 was between 24 and 36 h, the LT_50_ of strain 124 was between 72 and 96 h, and the LT50 of other strains was greater than 120 h. The larval survival rates of different isolates varied within 120 h after injection, ranging from 5.0% to 100.0% (Figure 1). Among the five isolates belonging to ST87 (112, 119, 121, 131, and 132), isolate 112 exhibited the lowest larval survival rate of only 16.7% at 120 h, which was equivalent to that of ATCC 19112 (Figure 1A). However, the remaining four isolates belonging to ST87 exhibited higher survival rates of 43.3% to 80.0% at 120 h (Figure 1A). The three isolates belonging to ST121 (127, 128, and 135) showed varying larval survival rates (Figure 1B). Among them, isolate 127 exhibited the lowest survival rate, decreasing to 10.0% at 12 h and reaching to 0% at 24 h (Figure 1B). Compared with ATCC 19112, isolate 128 displayed a lower survival rate (16.7%) within 12–120 h; however, isolate 135 exhibited a high larval survival rate of 70.0% after 120 h (Figure 1B). It was found that compared with ATCC 19112, the clinical strain 114 (ST1930) exhibited a lower larval survival rate, reaching only 10.0% after 24 h, while another ST1032 strain, 115, reached a level similar to that of ATCC 19112 (23.3%) after 120 h (Figure 1C). On the other hand, clinical isolates 113 and 116 demonstrated high larval survival rates, maintaining rates between 56.7% and 75.0% over a period of 120 h (Figure 1C).

The larval survival rates of ST8 (126 and 137) and ST9 (124, 125, and 134) isolates are shown in Figure 1D. Strain 125 exhibited a lower larval survival rate of 20.0% after 24 h compared with ATCC 19112. The larval survival rate of strain 124 was slightly higher than that of ATCC 19112 before 24 h, but the survival rate at 120 h remained as high as 43.3%. Additionally, isolates 126, 134, and 137 maintained high larval survival rates (70.0%, 76.7%, and 90.0%, respectively) during the 120 h. Furthermore, isolates belonging to ST451 (117 and 118) and ST5 (120 and 122) exhibited high larval survival rates, ranging from 46.7% to 86.7% at the end of 120 h (Figure 1E). Among other ST isolates, 136 (ST9*), 133 (ST307), 130 (ST155), 123 (ST2106), and 129 (ST3) also exhibited higher larval survival rates, ranging from 45.0% to 65.0% (Figure 1F).

The mortality rate of 26 *L. monocytogenes* isolates in *G. mellonella* larvae at 48 h was further graphed (Figure 2). The results showed that the mortality rate of isolates 127 (ST121), 128 (ST121), 114 (ST1930), and 125 (ST9) was significantly higher than that of other isolates (*p* < 0.05) after a period of 48 h (Figure 2A). Their mortality rates were 100.0%, 90.0%, 83.3%, and 80.0%, respectively (Figure 2A). According to ST classification, the mortality rate of ST121 (*n* = 3) was significantly higher than that of some other STs (ST87, ST451, ST5, ST8, ST2106, and ST3) (Figure 2B). The 48 h mortality rate of food-derived isolates was slightly higher than that of clinically derived isolates, but there was no significant difference (Figure 2C). The isolates from lineage II exhibited significantly higher mortality compared with those from lineage I (Figure 2D). There was no statistically significant difference in *G. mellonella* mortality among the different serogroups (Figure 2E). In this study, the injection dose was 10^4^ CFU per larva. Although at 48 h we could already distinguish the virulence differences between strains fairly well, we found that the LT_50_ for most strains was greater than 120 h. Therefore, in the future, if this injection concentration continues to be used, it will be necessary to further extend the culture time to obtain a more accurate LT_50_ for all strains for the comparison of strain virulence.

### 3.2. Cytotoxicity of Different L. monocytogenes Isolates

The cytotoxicity of *L. monocytogenes* to host cells was evaluated using human JEG-3 cells. As shown in Figure 3A, 26 *L. monocytogenes* isolates showed cytotoxicity toward JEG-3 monolayer cells. The cytotoxicity of isolates 121 (86.1%) and 115 (85.2%) was stronger than that of the positive control strain ATCC 19112 (78.9%). The cytotoxicity of the other strains ranged from 26.55% to 77.5%. Isolates 118 (26.55%), 134 (39.36%), 117 (40.56%), 112 (40.57%), 132 (43.44%), and 130 (45.16%) exhibited low cytotoxicity (<50%) (Figure 3A). According to ST classification, the isolates of ST451 (*n* = 2) exhibited significantly lower cytotoxicity compared with ATCC 19112 (ST122) and other STs (Figure 3B). Isolates belonging to ST1032 (*n* = 2) exhibited the highest cytotoxicity (Figure 3B). For overall cytotoxicity, clinically derived isolates were slightly more cytotoxic than food-derived isolates (Figure 3C). Lineage I exhibited higher values compared with lineage II (Figure 3D). Serogroup II b showed higher cytotoxicity than the other two serogroups (II a and II c) (Figure 3E). However, there were no significant differences in the cytotoxicity of *L. monocytogenes* based on strain origin, lineage, and serogroup classifications (*p* > 0.05) (Figure 3C–E).

### 3.3. Hemolytic Activity of 26 L. monocytogenes Isolates

The hemolytic activity of 26 strains of *L. monocytogenes* was investigated. Among these isolates, strain 112 (ST87) exhibited the highest hemolysis rate (33.43%), which was significantly higher than that of the standard strain ATCC 19112 (26.52%). The remaining isolates had hemolysis rates ranging from 26.52% to 32.29% (Figure 4A). Furthermore, there were no significant differences in hemolysis rate based on strain sources, lineages, and serogroups (Figure 4B–E).

### 3.4. Expression Levels of Key Virulence Genes in 26 Strains of L. monocytogenes

The expression of key virulence genes in *L. monocytogenes* determines the virulence level of a strain [1]. We employed RT-qPCR to examine the expression levels of six crucial virulence genes, including hemolysin (*hly*), internalin A (*inlA*), internalin B (*inlB*), positive regulatory factor A (*prfA*), phospholipase C A (*plcA*), and phospholipase C B (*plcB*) (Figure 5). Highly conserved codes for the Listeriolysin O (LLO) protein enable *L. monocytogenes* to escape from internalized host cell vacuoles via membrane perforation mechanisms [31]. InlA and InlB, encoded by the genes *inlA* and *inlB*, respectively, play a crucial role in the entry of *L. monocytogenes* into host cells [32]. The PrfA protein encoded by the *prfA* gene can regulate the transcription of most virulence genes during the process of infecting host cells, such as *hly*, *inlA*, *inlB*, *plcA*, *actA*, etc. [33]. The *plcA* and *plcB* genes, which encode phosphatidylinositol-specific phospholipase C (PI-PLC) and phosphatidylcholine-specific phospholipase C (PC-PLC), respectively, are crucial for bacteria to escape from the phagosomes [34]. In our results, except for strain 114, which showed no significant difference in *hly* gene expression level compared with the standard strain ATCC 19112 (*p* > 0.05), the *hly* gene expression levels of the other 25 isolates were significantly lower than that of ATCC 19112 (*p* < 0.05) (Figure 5A). *inlA* expression in isolates 127, 128, and 135 was significantly higher than in the other 24 strains, including ATCC 19112 (*p* < 0.05) (Figure 5B). The expression levels of *inlB* in isolates 114, 127, 128, and ATCC 19112 were significantly higher than those in the other 23 strains (*p* < 0.05) (Figure 5C). Among the 26 strains of *L. monocytogenes*, only strain 128 exhibited significantly higher *prfA* gene expression compared with other isolates (*p* < 0.05) (Figure 5D). The results for the *plcB* gene were highly similar to those for *plcA*. Only 128 isolates showed significantly higher expression of both the *plcA* and *plcB* genes compared with the other strains (*p* < 0.05) (Figure 5E,F).

The expression results for the *hly*, *inlA*, *inlB*, *prfA*, *plcA*, and *plcB* genes were classified and reanalyzed at the levels of ST, strain source, lineage, and serogroup. The result showed that *hly* gene expression levels were significantly higher in the ST121 (*n* = 3) and ST155 (*n* = 1) types compared with in other STs, except for the ST1930 type (*p* < 0.05) (Figure 6A). No significant differences were observed in the relative expression of *hly* among different strain sources or serogroups (Figure 6A). The *hly* gene expression of isolates belonging to lineage II showed a higher trend than that of isolates belonging to lineage I, but the difference was not statistically significant (*p* = 0.069) (Figure 6A). The classification and reanalysis of *inlA* gene expression also revealed that ST121 (*n* = 3) exhibited significantly higher gene expression compared with other STs (*p* < 0.05) (Figure 6B). The expression of the *inlA* gene in food-derived isolates, including ST121, was significantly higher than that observed in clinical isolates (*p* < 0.01) (Figure 6B). The *inlA* expression of isolates belonging to lineage II, including ST121, was significantly higher than that of isolates belonging to lineage I (*p* < 0.01) (Figure 6B). Furthermore, based on serogroup classification, group II a exhibited significantly higher expression of the *inlA* gene compared with groups II b (*p* < 0.05) and II c (*p* < 0.05) (Figure 6B). ST121 (*n* = 3), ST1930 (*n* = 2), and ST122 (ATCC 19112) had significantly higher *inlB* gene expression levels than other STs (*p* < 0.05) (Figure 6C). There were no significant differences in *inlB* expression based on strain origin or lineage (Figure 6C). For different serogroups, group II a showed significantly higher expression of the *inlB* gene than group II c (*p* < 0.05) (Figure 6C).

After classification and reanalysis, it was found that ST121, containing strain 128, showed significantly higher *prfA* gene expression levels than other STs (*p* < 0.05) (Figure 6D). The expression of *prfA* in food-isolated strains was significantly higher than that in clinically isolated strains (Figure 6D). There was no statistically significant difference in *prfA* expression among different lineages (*p* > 0.05) (Figure 6D). For different serogroups, group II c showed significantly higher expression of the *prfA* gene than group II a (*p* < 0.05) (Figure 6D). Similarly, when the results were classified according to ST, the *plcA* gene expression of ST121 (*n* = 3), including strain 128, was significantly higher than that of other STs (*p* < 0.05) (Figure 6E). When results were classified according to strain source or lineage, *plcA* expression in food-derived isolates (excluding strain 128) was significantly higher than that in clinical isolates (*p* < 0.05) (Figure 6E). The *plcA* expression of lineage II isolates (excluding 128) was significantly higher than that of lineage I (*p* < 0.01) (Figure 6E). The expression of *plcA* in serogroup II a was significantly higher than that in serogroup II b (*p* < 0.05) (Figure 6E). Additionally, the expression of the *plcB* gene in ST121 (including 128) was higher than that in other STs (Figure 6F). The *plcB* expression of food-isolated strains was significantly higher than that of clinically isolated strains (*p* < 0.01) (Figure 6F). The isolates from lineage II exhibited significantly higher *plcB* expression compared with those from lineage I (*p* < 0.01) (Figure 6F). In addition, *plcB* expression in serogroups II a and II c was significantly higher than that in serotype II b (*p* < 0.05) (Figure 6F).

### 3.5. Correlation Analysis of Factors Affecting L. monocytogenes Virulence

We conducted gray relational analysis (GRA) based on the aforementioned virulence indicators and drew a gray correlation heat map (Figure 7). The results showed that the pathogenicity of *L. monocytogenes* was most correlated with the expression of *inlB*, followed by *hly*. In addition, the correlation between hemolytic activity and cytotoxicity was the strongest, and the correlation index (CI) was 0.950. The expression of *inlA* and *inlB* was closely related (CI = 0.881). The correlation between *hly* and *prfA* expression was stronger compared with other factors (CI = 0.861). *plcA* expression also had a strong relationship with *plcB* expression (CI = 0.890). Furthermore, the relative expression level of the *prfA* gene was strongly related with hemolytic activity (CI = 0.872) and cytotoxicity (CI = 0.868), respectively. However, the gray CI between the relative expression level of the *hly* gene and hemolytic activity was 0.849.

## 4. Discussion

In this study, the virulence of 26 strains of *L. monocytogenes* isolated from food and clinical samples in Shanghai, China, from 2020 to 2022 was investigated, including their mortality rate in *G. mellonella*, cytotoxicity to human cells, hemolytic activity, and expression of important virulence genes. In recent years, *Galleria mellonella* larvae have been widely used as an alternative mammalian model for evaluating the pathogenesis of many human pathogens [35,36,37]. The *G. mellonella* model has shown a good correlation with mammalian models in the study of the virulence of *L. monocytogenes* strains, primarily due to its similarity to mammals in terms of the immune response system, resistance against pathogens (based on lysozymes, reactive oxygen species, and antimicrobial peptides), and the ability to live at 37 °C, which is suitable for human pathogens [38]. A study on *G. mellonella* infection reported that among the different serotypes tested, strains of serotypes 4b and 1/2a (EGD-e) were more pathogenic than other serotypes of *L. monocytogenes*, such as 4a, 4c, and 4d [38]. In this study, 4b serogroup strains were not used; however, the 48 h mortality in serogroup II a was significantly higher than that in II b and II c. Another report reveals that 11 out of 13 clinical isolates from past listeriosis outbreaks showed significantly higher virulence potential compared to isolates from foods associated with the same listeriosis outbreaks in an inoculum of 10^6^ CFU/larva [26]. Due to the difficulty in food traceability caused by sporadic clinical samples, we were unable to obtain both food and clinical samples from the same clinical event in this study. Among the 26 strains, there was no significant difference in *G. mellonella* mortality between clinical and food source bacteria.

A systematic review of *L. monocytogenes* in China from 2010 to 2019 revealed that the proportion of ST121 among clinical isolates was 4.81% (5/104), while it accounted for 4.76% (57/1197) in food samples [2]. Although ST121 was primarily detected in food, it is also capable of causing clinical symptoms. Our results showed that the ST121 type, including strains 125, 127, and 128, exhibited a high mortality rate in the *G. mellonella* model, indicating potentially high pathogenicity in humans. In addition, Cheng et al.’s ten-year meta-analysis also showed that ST87 (15.38%, 16/104), ST8 (13.46%, 14/104), and ST5 (11.54%, 12/104) were considered the most common clinical STs in China [2]. Some scholars believe that strains causing a high frequency of occurrence in invasive listeriosis are also to be classified as highly virulent strain groups [39]. For instance, ST87 exhibits a high clinical detection rate, classifying it as a highly virulent group, whereas ST5 and ST8 are considered moderately virulent groups. Conversely, ST121 and ST9 are regarded as low-virulence groups [39]. However, our results showed that ST121 and ST9 exhibited higher virulence compared with other STs based on the *G. mellonella* model. Our cytotoxicity experiments based on JEG-3 cells showed no significant differences among ST8, ST9, ST121, and ST87. The results indicated a significant difference between *in vitro* human cell experiments and *in vivo G. mellonella* toxicity experiments when the virulence of various ST strains was assessed. Some studies suggest that *G. mellonella* could effectively differentiate the virulence of *L. monocytogenes* [26,40]. However, a comparison of the two models revealed that the results of virulence differences for different STs from the JEG-3 cell model more closely resemble statistical results for the frequency of human listeriosis caused by strains of different STs [39]. Consequently, the JEG-3 cell model might be a valuable tool for evaluating the virulence of various STs of *L. monocytogenes* in the future. However, since this model is based on a single type of human tumor-derived cell, it cannot fully replicate actual human *in vivo* outcomes. Moving forward, conducting toxicity analyses with organoids or multi-subtype 3D cell models created from human cells might provide a more accurate assessment of strain virulence. In the future, it will still be necessary to verify the differences in virulence between different ST strains from multiple perspectives, such as whole-genome sequencing information, adhesion and invasion experiments on human cells, or mammalian organ burden experiments, in order to explain the results of clinical incidence accurately. 

*Listeria* pathogenicity Island 1 (LIPI-1) contains virulence genes necessary for determining the pathogenicity of *L. monocytogenes*, particularly *hly*, which encodes porin LLO [41]. Since all strains in this study had intact *hly* sequences, we additionally evaluated if the transcriptional expression and hemolytic activity of *hly* played a significant role in determining their virulence. Previous studies have shown that *L. monocytogenes* responds to infection in *G. mellonella* larval hosts by significantly inducing the expression of virulence genes and reducing the viability of blood cells [40]. Compared with the wild-type *L. monocytogenes* strain (EGDe), the *hly* mutant (EGDeΔ*hly*) did not cause mortality in *G. mellonella* larvae [40]. Our results showed that the relative expression level of *hly* was found to be higher in isolates belonging to lineage II compared to those belonging to lineage I (*p* = 0.069). Additionally, isolates belonging to ST121 and ST1930 exhibited higher *hly* gene expression. This was consistent with the 48 h mortality rate in *G. mellonella* infected by *L. monocytogenes*. These findings confirm that *hly* plays a crucial role in the assessment of the *in vivo* virulence of *L. monocytogenes*.

Our results showed that some STs, such as ST1032, ST5, and ST3, which had high cytotoxicity in placental cells, did not exhibit high *hly* gene expression. Furthermore, our gray-scale correlation analysis revealed a strong correlation between hemolysis activity and cytotoxicity compared with the other virulence assessment factors of strains. This suggests that the mechanism of virulence acting on *in vitro* cell models is highly similar to the hemolysis effect of LLO. This may be due to the fact that *L. monocytogenes* has different infection mechanisms in human cells and in *G. mellonella*, with *hly* being involved in infecting both. For human cells, the secretion of LLO by *L. monocytogenes* can facilitate bacterial internalization by creating pores in the plasma membrane, leading to endocytosis into epithelial cells or hepatocytes [42]. After *L. monocytogenes* internalizes into host cells, LLO plays a role in its main action site (phagocytic membrane) and causes damage to the lipid membrane, resulting in toxic effects on cells [43]. On the other hand, it has been reported that the loss of hemocytes may be associated with the death of *G. mellonella* and is driven by LLO expression [40]. Further research is necessary to demonstrate mechanisms, but these results suggest that *L. monocytogenes* with higher *hly* gene expression exhibits stronger *in vivo* virulence and pathogenicity toward *G. mellonella*.

The surface proteins InlA and InlB are described as the main invasive proteins responsible for the uptake of *L. monocytogenes* by normal non-phagocytic cells [44]. Compared with the wild-type strain (EGDe), the *prfA* mutant (EGDeΔ*prfA*) prolonged the LT_50_ (lethal time of 50%) of *G. mellonella*, while mutations in *inlA* (EGDeΔ*inlA*) and the mutation in *inlB* (EGDeΔ*inlB*) did not affect LT_50_ [40]. Despite both *inlA* and *inlB* being internalin proteins, they exhibit distinct mechanisms of action. The InlA protein mediates bacterial entry only into cells expressing E-cadherin, while InlB is a more versatile invasive protein due to its receptors being widely expressed [44]. The N-terminal leucine-rich repeat (LRR) domain of InlB binds to c-Met (or the HGF receptor), and the C-terminal portion of InlB also binds to glycosaminoglycans and gC1Q receptors, in addition to anchoring lipoteichoic acid [45]. The *inlA* mutants of strain 10423S have been reported to exhibit a toxicity consistent with the wild-type strain after direct injection into the blood cavity of *G. mellonella* larvae [46]. It is hypothesized that, similarly to the mouse model, this may be due to the absence of E-cadherin in *G. mellonella* larvae, which specifically binds to InlA. Our results showed that the expression patterns of *inlA* and *inlB* were not completely identical. For example, strains 127 (ST121) and 128 (ST121) exhibited simultaneous high expression of both *inlA* and *inlB*, whereas strain 114 (ST1930) only showed high expression of *inlB*, while strain 135 (ST121) only exhibited high expression of *inlA*. However, in the experiment on *G. mellonella*, only strains 127, 128, and 114 exhibited high mortality rates. This result also suggests a close relationship between the high expression of *inlB* and the mortality of *G. mellonella*, which was confirmed through a gray relation analysis of virulence factors.

The PrfA protein is necessary for the activation of operons in LIPI-1, and it is selectively activated during host cell infection, promoting the expression of *hly*, *inlA*, *inlB*, *inlC actA*, *plcA*, and *plcB* [47]. The first gene of the *plcA*-*prfA* operon encodes PlcA, which collaborates with PlcB and LLO to destroy primary vacuoles formed after the phagocytosis of *L. monocytogenes* extracellularly [41]. In this study, RT-qPCR results showed that the gene expression patterns of both *plcA* and *plcB* were consistent with those of *prfA*. Isolate 128, which had a significant gene expression advantage in *plcA*, *plcB*, and *prfA*, also demonstrated significantly higher mortality in the *in vivo* model of *G. mellonella* larvae.

## 5. Conclusions

In this research, we examined the virulence of different *L. monocytogenes* strains, including their mortality rate in *G. mellonella*, cytotoxicity to human JEG-3 cells, hemolytic activity, and expression of important virulence genes. The results showed that there were differences in virulence between different strains or different STs. The mortality rate in *G. mellonella* was significantly higher for lineage II compared to lineage I. The isolates belonging to ST121 and ST1930 strains exhibited significantly higher mortality rates. There was a strong correlation between mortality in *G. mellonella* and the expression of the *inlB* and *hly* genes. The toxicity pattern for *G. mellonella in vivo* differed from the cytotoxicity observed in JEG-3 cells *in vitro*. Compared with other STs, ST1930, ST3, ST5, and ST1032 exhibited higher cytotoxicity toward JEG-3 cells. There was a strong correlation between hemolytic activity and cell toxicity among different strains. Compared with clinical strains, foodborne strains exhibited higher expression of *inlA*, *prfA*, *plcA*, and *plcB*. However, there were no significant differences observed between the two sources in terms of cytotoxicity, *G. mellonella* mortality, and hemolytic activity. Although some important conclusions about the prevalent ST were obtained in this study, it is necessary to increase the number of isolates with the same ST in future studies to ensure data balance and obtain more reliable results. Due to the inconsistency between the evaluations of cell models and insect models in this study, further measurements of virulence levels in *in vivo* and *in vitro* models will be beneficial for a better understanding of the detailed virulence mechanism of *L. monocytogenes* using the 26 isolates in the future. Additionally, conducting genome-wide association studies (GWASs) is essential for further proving the correlation between phenotype and genotype.

## Figures and Tables

**Figure 1 microorganisms-13-00191-f001:**
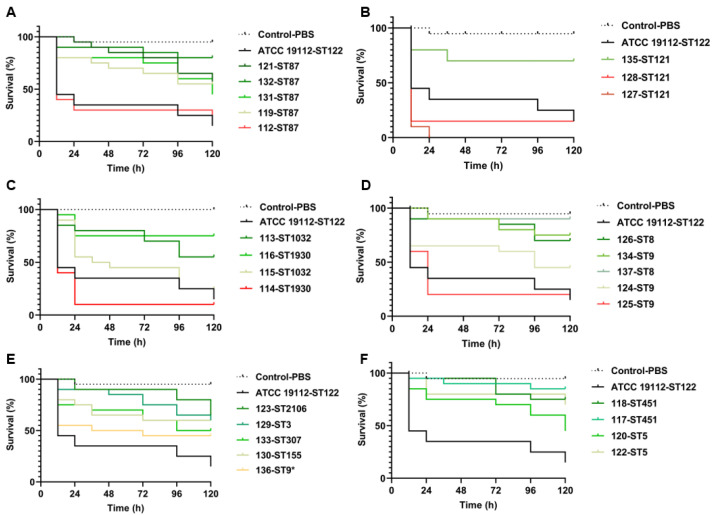
The survival rate of 26 *L. monocytogenes* strains over 120 h based on an infection model of *G. mellonella* larvae. Survival curves of *G. mellonella* larvae infected with ST87 strains (112, 119, 121, 131, and 132) (**A**), ST121 strains (127, 128, and 135) (**B**), ST1032 and ST1930 strains (**C**), ST8 and ST9 strains (**D**), ST2106, ST3, ST307, ST155, and ST9* strains (**E**), ST451 and ST5 strains (**F**).

**Figure 2 microorganisms-13-00191-f002:**
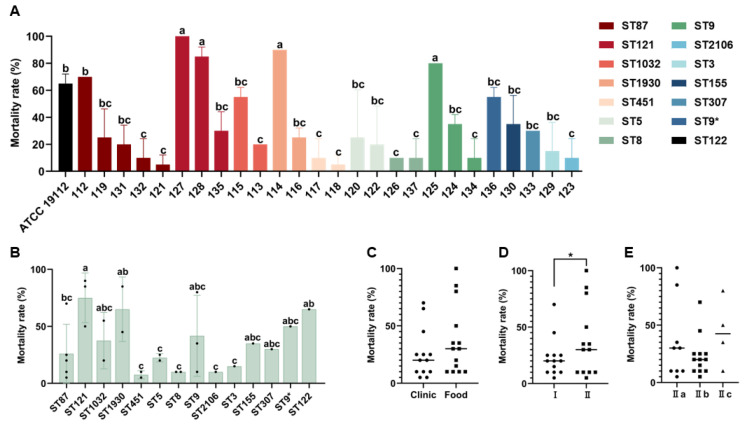
The 48 h mortality of 26 *L. monocytogenes* isolates in the larvae of *G. mellonella* based on different classifications, including strain levels (**A**), STs (**B**), strain sources (The circles represents clinical strains, and the squares represents food strains) (**C**), lineages (The circles represents lineage I strains, and the squares represents lineage II strains) (**D**), and serogroups (The circles represents serotype II a strains, the squares represents serotype IIb strains, and the triangles represents serotype II c strains) (**E**). * Represent a mutation in the base sequence at ST9. Since not all STs have more than three strains, the analysis of significant differences was conducted for STs with only one or two strains, based on the values obtained from three repeated experiments of that particular strain. Different lowercase letters (a~c) indicate significant differences (*p* < 0.05). * *p* < 0.05 indicates a significant difference between groups.

**Figure 3 microorganisms-13-00191-f003:**
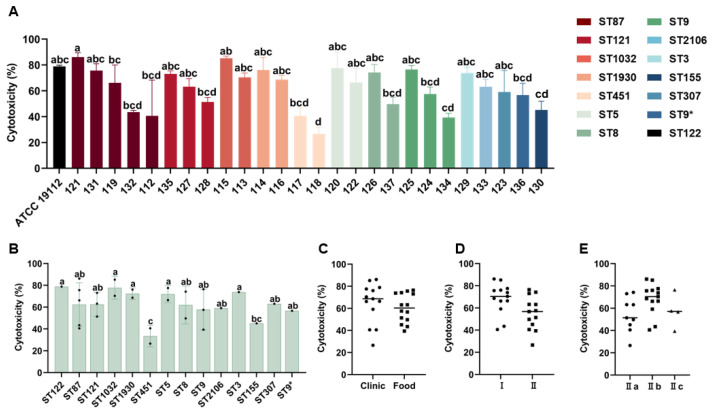
Differences in cytotoxicity among 26 strains of *L. monocytogenes* toward JEG-3 cells. *Listeria monocytogenes* was added into JEG-3 monolayer cells and incubated at 37 °C for 24 h. The cytotoxicity of 26 strains of *L. monocytogenes* to JEG-3 cells based on (**A**) different STs (**B**), strain sources (**C**) (The circles represents clinical strains, and the squares represents food strains), lineages (**D**) (The circles represents lineage I strains, and the squares represents lineage II strains), and serogroups (The circles represents serotype II a strains, the squares represents serotype IIb strains, and the triangles represents serotype II c strains) (**E**) was observed. * Represent a mutation in the base sequence at ST9. Since not all STs have more than three strains, the analysis of significant differences was conducted for STs with only one or two strains, based on the values obtained from three repeated experiments for that particular strain. Different lowercase letters (a~d) indicate significant differences (*p* < 0.05).

**Figure 4 microorganisms-13-00191-f004:**
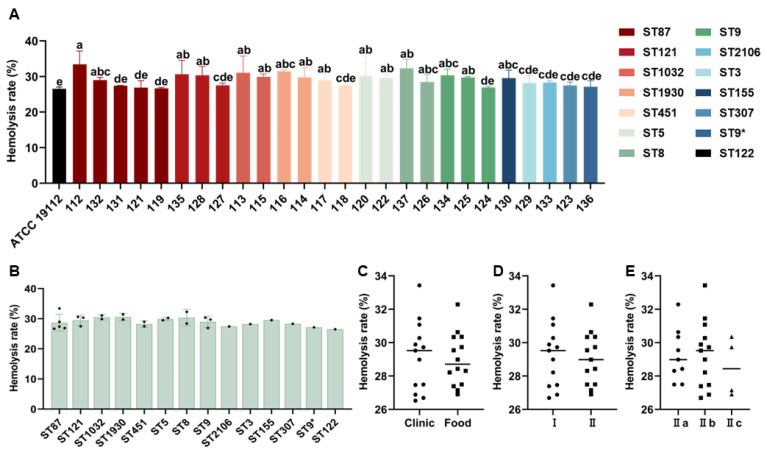
The hemolytic activity of 26 strains of *L. monocytogenes* (**A**). The difference in the hemolysis rate of *L. monocytogenes* was observed based on different STs (**B**), strain sources (The circles represents clinical strains, and the squares represents food strains) (**C**), lineages (The circles represents lineage I strains, and the squares represents lineage II strains) (**D**), and serogroups (The circles represents serotype II a strains, the squares represents serotype IIb strains, and the triangles represents serotype II c strains) (**E**). * Represent a mutation in the base sequence at ST9. Since not all STs have more than three strains, the analysis of significant differences was conducted for STs with only one or two strains, based on the values obtained from three repeated experiments for that particular strain. Different lowercase letters (a–e) indicate significant differences (*p* < 0.05).

**Figure 5 microorganisms-13-00191-f005:**
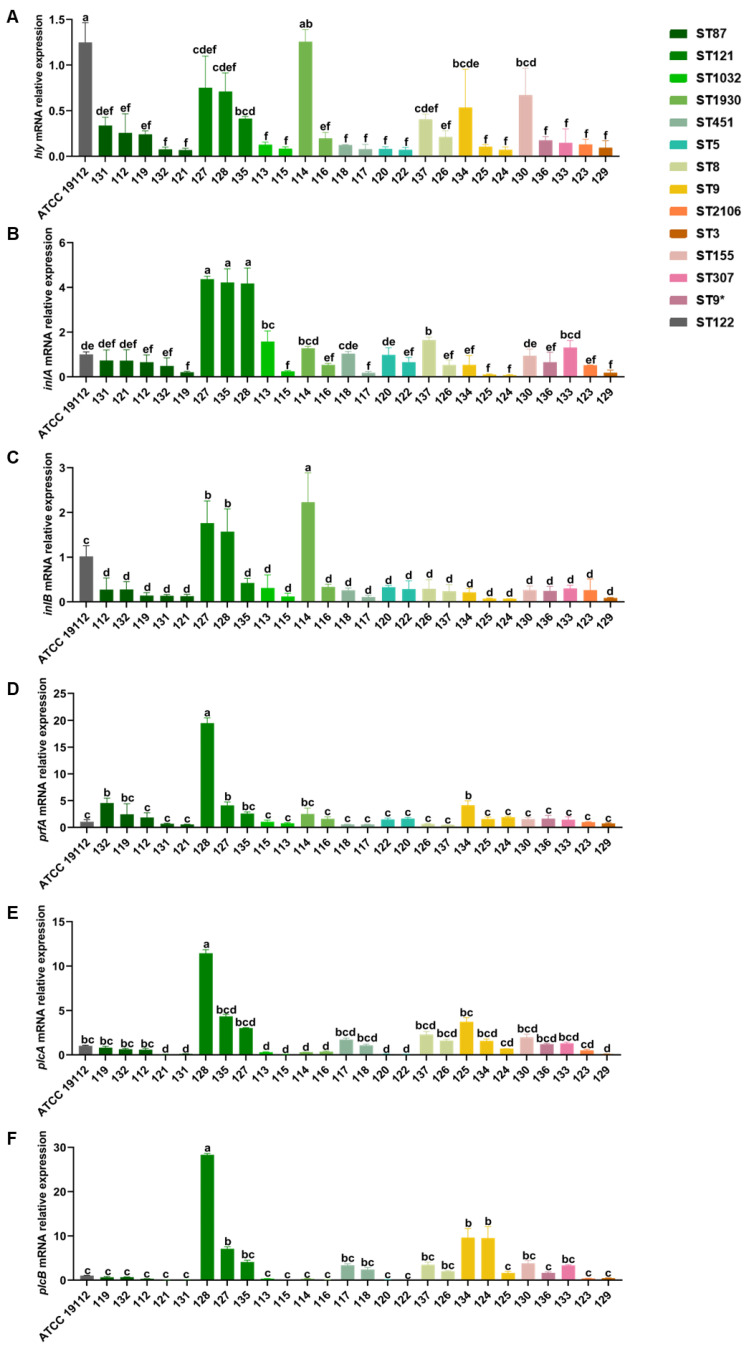
The relative expression levels of key virulence genes *hly* (**A**), *inlA* (**B**), *inlB* (**C**), *prfA* (**D**), *plcA* (**E**), and *plcB* (**F**) among 26 *L. monocytogenes* isolates. Different lowercase letters (a~f) indicate significant differences (*p* < 0.05). * Represent a mutation in the base sequence at ST9.

**Figure 6 microorganisms-13-00191-f006:**
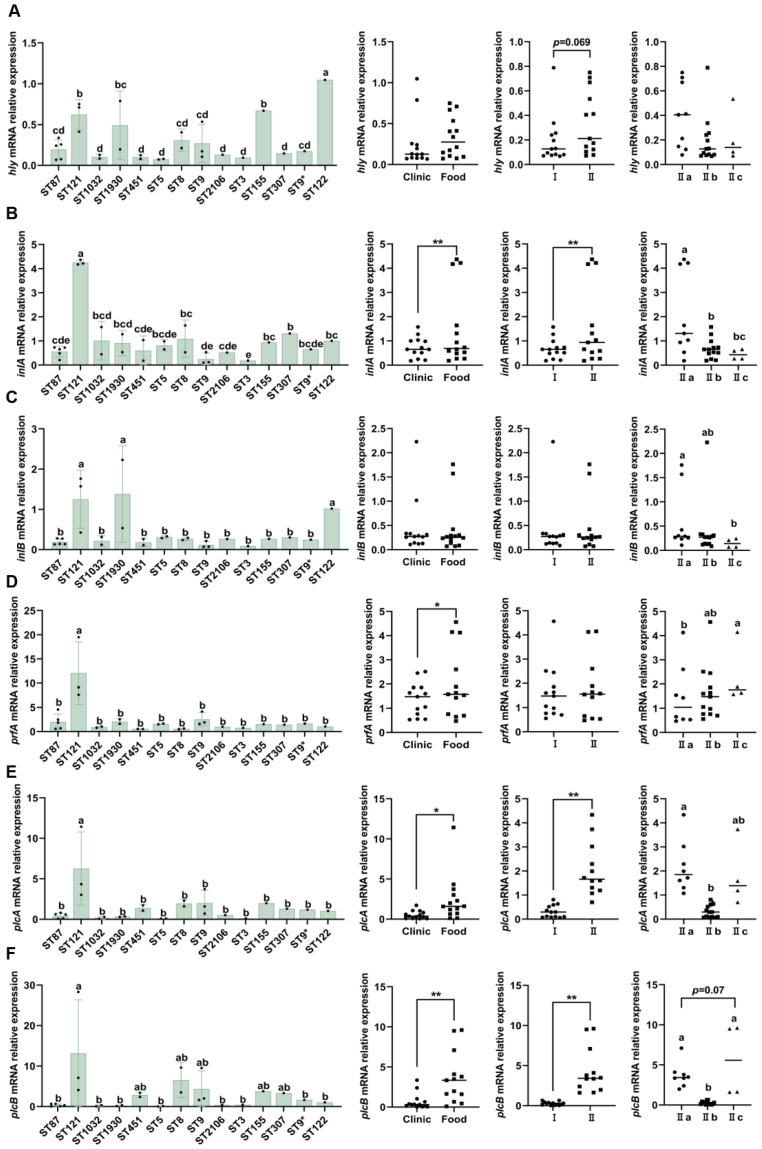
The differences in the expression of genes *hly* (**A**), *inlA* (**B**), *inlB* (**C**), *prfA* (**D**), *plcA* (**E**), and *plcB* (**F**) based on subtype classification of the ST, strain source (The circles represents clinical strains, and the squares represents food strains), lineage (The circles represents lineage I strains, and the squares represents lineage II strains), and serogroup (The circles represents serotype II a strains, the squares represents serotype IIb strains, and the triangles represents serotype II c strains). Since not all STs have more than three strains, the analysis of significant differences was conducted for STs with only one or two strains, based on the values obtained from three repeated experiments for that particular strain. In (**D**–**F**), the data sets derived from food source, lineage II, and serotype II a do not conform to a normal distribution. Therefore, taking into consideration the idea that strain 128 may be an outlier, (**D**–**F**) display the results of re-performing Student’s *t*-test after excluding 128 data points. Different lowercase letters (a~e) indicate significant differences (*p* < 0.05). * *p* < 0.05 and ** *p* < 0.01 indicate a significant difference between groups.

**Figure 7 microorganisms-13-00191-f007:**
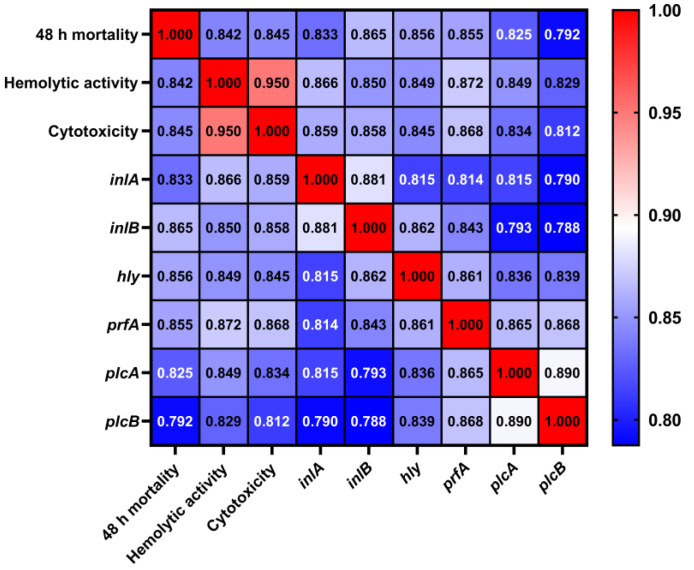
Gray relation heat map of factors related to the virulence of *L. monocytogenes*. The higher the gray correlation coefficient (approaching red), the stronger the correlation between the two factors.

**Table 1 microorganisms-13-00191-t001:** RT-PCR primer name and sequence.

Gene	Primer	Sequence (5′-3′)
*16S*	forward	ACATCCTTTGACCACTCTGGA
	reverse	CAACATCTCACGACACGAGC
*inlA*	forward	ATAGGCACATTGGCGAGTTT
	reverse	GTGCGGTTAAACCTGCTAGG
*inlB*	forward	AAGCAMGATTTCATGGGAGAGT
	reverse	TTACCGTTCCATCAACATCATAACTT
*actA*	forward	CGGGTAAATGGGTACGTGAT
	reverse	TGGTCAATTAACCCTGCACTT
*prfA*	forward	CAACATCTCACGACACGAGC
	reverse	GCTAACAGCTGAGCTATGTGC
*sigB*	forward	TCATCGGTGTCACGGAAGAA
	reverse	TGACGTTGGATTCTAGACAC
*hly*	forward	CTTTTAACCGGGAAACACCA
	reverse	TCTTGCGTTACCTGGCAAA

**Table 2 microorganisms-13-00191-t002:** LT50 after 26 strains of *L. monocytogenes* were injected into the larvae of *G. mellonella*.

Strain Number	Serogroup *	Lineage *	ST *	Source	Year	Location	LT_50_
112	II b	I	ST87	Human blood	2021	Shanghai, China	<12 h
113	II b	I	ST1032	Human blood	2021	Shanghai, China	>120 h
114	II b	I	ST1930	Human blood	2021	Shanghai, China	<12 h
115	II b	I	ST1032	Human uterine contents	2021	Shanghai, China	≤36 h
116	II b	I	ST1930	Human uterine contents	2021	Shanghai, China	>120 h
117	II a	II	ST451	Human uterine contents	2021	Shanghai, China	>120 h
118	II a	II	ST451	Human blood	2021	Shanghai, China	>120 h
119	II b	I	ST87	Human blood	2021	Shanghai, China	>120 h
120	II b	I	ST5	Human blood	2021	Shanghai, China	<120 h
121	II b	I	ST87	Human blood	2022	Shanghai, China	>120 h
122	II b	I	ST5	Human blood	2022	Shanghai, China	>120 h
123	II b	I	ST2106	Human vaginal secretion	2022	Shanghai, China	>120 h
124	II c	II	ST9	Chilled pork	2021	Shanghai, China	<96 h
125	II c	II	ST9	Fresh pork	2021	Shanghai, China	≤24 h
126	II a	II	ST8	Frozen processed beef	2021	Shanghai, China	>120 h
127	II a	II	ST121	Frozen processed beef	2021	Shanghai, China	<12 h
128	II a	II	ST121	Chilled chicken	2021	Shanghai, China	<12 h
129	II b	I	ST3	Chilled duck	2021	Shanghai, China	>120 h
130	II a	II	ST155	Chilled chicken	2021	Shanghai, China	>120 h
131	II b	I	ST87	Cooked beef	2021	Shanghai, China	>120 h
132	II b	I	ST87	Chilled cooked beef	2021	Shanghai, China	>120 h
133	II a	II	ST307	Chilled cooked beef	2021	Shanghai, China	>120 h
134	II c	II	ST9	Frozen processed fork	2020	Shanghai, China	>120 h
135	II a	II	ST121	Chilled processed chicken	2020	Shanghai, China	>120 h
136	II c	II	ST9	Frozen processed chicken	2020	Shanghai, China	≤36 h
137	II a	II	ST8	Frozen processed beef	2020	Shanghai, China	>120 h

* The ST, serogroup, and lineage in Table 2 are determined based on the whole-genome sequencing results of the strains.

## Data Availability

The original contributions presented in this study are included in this article, and further inquiries can be directed to the corresponding author.

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
