# Peer review of "Comparative Analysis of In Vivo and In Vitro Virulence Among Foodborne and Clinical Listeria monocytogenes Strains"

_microorganisms, 2025, doi:10.3390/microorganisms13010191_

Round 1
Reviewer 1 Report
Comments and Suggestions for Authors
In their manuscript, the authors present the results regarding the virulence of different L. monocytogenes strains on G. mellonella, cytotoxicity to human JEG-3 cells, hemolytic activity, and expression of important virulence genes. For this aim, they used 26 strains of L. monocytogenes isolated from China, 12 from clinical sources and 14 from food samples. The results showed that there are differences in virulence between different strains. They found a strong correlation between the mortality of G. mellonella and the expression of inlB and hly genes and a strong correlation between hemolytic activity and cell toxicity among different strains.
This manuscript's novelty consists of the model adopted for virulence study, developed on G. mellonella larvae as an alternative to the lab animal model. The authors explained with scientific arguments their choice: the species G. mellonella gave good correlations with mammalian models in the studies performed on the virulence of L. monocytogenes due to its similarity to mammals in terms of the immune response system, resistance against pathogens (based on lysozyme, reactive oxygen species, and antimicrobial peptides), and ability to survive at 37°C). The manuscript is well written, experiments were conducted logically, the methodology used is correct, and the conclusion sustains the results obtained. For these reasons, I recommend publishing.
Minor corrections:
1) A sentence cannot begin with an abbreviation. Authors must read them with attention to their manuscript and rewrite all these sentences ( see the sentences from the following rows: R45; R 144, R 147, R 287, R291, etc.)
2) The manuscript body text must be formatted according to MDPI rules ( the font size appears to not be the same in the manuscript; between the sentences exist extra spaces). This issue will be solved by the authors or by the editor)
3) Some abbreviations are not explained in the text ( for example, LT 50; delta inlB ( see the R 454) ) These must explained in the manuscript, in the brackets, the first time when they are mentioned in the text.
4) The pictures and the tables must appear in the manuscript on the same page in which they are mentioned.
Reviewer 2 Report
Comments and Suggestions for Authors
Dear authors, the paper is a relevant subject with merit to be published, however, I recommend a review for the following observations:
Suggested keywords: Listeria monocytogenes; Virulence; hemolysis activity; RT-PCR, Galleria mellonella, JEG-3 cells.
The samples were isolated in the same period. You mentioned in the discussion that there was no relationship between the clinical isolates and the food samples. How was this analysis performed?
Were these samples typed using the MLST technique or sequenced?
What were the food and clinical sources?
I suggest including, in supplementary material, a table with the raw data of the samples: clinical/food samples, year or month, location, source of isolation, ST, virulence results.
In topic 2.2, line 117, what was the inoculated quantity of Galleria mellonella larvae? Did you base this on any reference?
What were the references in topics 2.3, 2.4 and 2.5?
Two infection periods were evaluated, 48h and 120h. Which would you choose as ideal for further testing?
Regarding cytotoxicity analyses for G. mellonella and in human JEG-3 cells, which would you choose for future testing?
